# Quality assessment of Second-generation Global Imager (SGLI)-observed cloud properties using SKYNET surface observation data

Pradeep Khatri[1], Tadahiro Hayasaka[1], Hitoshi Irie[2], Husi Letu[3], Takashi Y. Nakajima[4], Hiroshi Ishimoto[5], and Tamio Takamura[2]

[1]Center for Atmospheric and Oceanic Studies, Graduate School of Science, Tohoku University, Sendai, Japan

[2]Center for Environmental Remote Sensing, Chiba University, Chiba, Japan

[3]Institute of Remote Sensing and Digital Earth, Chinese Academy of Sciences, Beijing, China

[4] Research and Information Center (TRIC), Tokai University, Hiratsuka, Japan

[5]Meteorlogical Research Institute, Tsukuba, Japan

*Correspondence to*: Pradeep Khatri (pradeep.khatri.a3@tohoku.ac.jp)

**Abstract.** The Second-generation Global Imager (SGLI) onboard the Global Change Observation Mission – Climate (GCOM-C) satellite launched on December 23, 2017, observes various geophysical parameters with the aim of a better understanding of the global climate system. As part of that aim, SGLI has great potential to unravel several uncertainties related to clouds by providing new cloud products along with several other atmospheric products related to cloud climatology, including aerosol products from polarization channels. However, a very little is known about the quality of the SGLI cloud products. This study uses data about clouds and global irradiances observed from the Earth's surface using a sky radiometer and a pyranometer, respectively, to understand the quality of the two most fundamental cloud properties—cloud optical depth (COD) and cloud-particle effective radius (CER)—of both water and ice clouds. The SGLI-observed COD agrees well with values observed from the surface, although it agrees better for water clouds than for ice clouds, while the SGLI-observed CER exhibits poorer agreement than does the COD, with the SGLI values being generally higher than the sky radiometer values. These comparisons between the SGLI and sky radiometer cloud properties are found to differ for different cloud types of both the water and ice cloud phases and different solar and satellite viewing angles by agreeing better for relatively uniform and flat cloud type and for relatively low solar zenith angle. Analyses of SGLI-observed reflectance functions and values calculated by assuming plane-parallel cloud layers suggest that SGLI-retrieved cloud properties can have biases on the solar and satellite viewing angles, similar to other satellite sensors including the Moderate Resolution Imaging Spectroradiometer (MODIS). Furthermore, it is found that the SGLI-observed cloud properties reproduce global irradiances quite satisfactorily for both water and ice clouds by resembling several important features of the COD comparison, such as the better agreement for water clouds than

for ice clouds and the tendency to underestimate (resp. overestimate) the COD in SGLI observations for optically
thick (resp. thin) clouds.

## 1. Introduction

Clouds play important roles in changing the Earth's climate system (Ramanathan et al., 1989), with profound
impacts on the atmospheric heat budget and the hydrological cycle (Rosenfeld et al., 2014). However, their strong
spatial and temporal variations as well as their complex interactions with aerosols and meteorology (e.g., Rosenfeld
et al., 2014; Khatri et al., 2020) have made it difficult to date to represent clouds accurately in global climate
models (Forster et al., 2021). Consequently, the roles of clouds in climate change are very poorly understood, and
they are highlighted as important sources of uncertainty in future climate projections (IPCC, 2021). Given their
importance, clouds are now being studied from different perspectives and using different methods, one of which
is cloud remote sensing from space, which has been in practice since the first successful capture of a cloud picture
by a Television InfraRed Observational Satellite (TIROS) launched on April 1, 1960. Since then, cloud remote-
sensing technology has advanced greatly, and there are currently several active and passive sensors onboard
various polar-orbiting or geostationary satellites to observe clouds from space. Because of their benefits of wide
spatial coverage and continuous observations at specific time intervals, satellite cloud products have been used
broadly either independently (e.g., Khatri et al., 2021) or combined with technologies such as numerical simulation
or artificial intelligence (e.g., Masunaga et al., 2008; Letu et al., 2020, 2021), for a better understanding of cloud
climatology as well as energy and water budgets. However, because the same satellite sensor monitors either the
whole Earth or a large part of it for a long time and the cloud products are generally generated by processing
satellite-received signals using certain physical models and assumptions (e.g., daytime cloud optical depth (COD)
and cloud-particle effective radius (CER) are obtained using the reflectance observed at two different wavelengths
by assuming clouds to be plane-parallel horizontal (PPH) layers), assessing the quality of such cloud products is a
fundamental requirement for using them in scientific research, policy making, and other application areas.
Furthermore, such quality-assessment studies help in gathering important information that is useful for developing
next-generation satellite sensors and observation techniques that overcome the shortcomings of existing
technologies.
The Global Change Observation Mission – Climate (GCOM-C) satellite (or "Shikisai" in Japanese) is a polar-
orbiting satellite that was launched on December 23, 2017. Onboard is the Second-generation Global Imager
(SGLI), which has 16 channels covering the spectrum from ultraviolet to thermal infrared. Of these 16 channels,
the 1.05-, 1.63-, and 2.21-μm channels in the shortwave infrared region and the 10.8-μm channel in the thermal
region are used to infer the properties of both water and ice clouds (Nakajima et al., 2019). Having entered
operation relatively recently, very little is known about the quality of the cloud products generated from the SGLI
satellite sensor, thereby emphasizing the need and urgency for assessing the quality of SGLI cloud products. In
addition, SGLI is a powerful sensor for observing aerosols because of the inclusion of polarization and
bidirectional channels, thereby making it very useful for studying aerosol–cloud interaction with qualitative
aerosol data. Therefore, studies related to assessing the quality of SGLI cloud products can also contribute to
aerosol–cloud interaction studies performed using SGLI data.
A literature review shows the scarcity of quality-assessment studies for SGLI cloud products. Nakajima et al.
(2019) performed such a study by comparing SGLI cloud products with those obtained from the Moderate
Resolution Imaging Spectroradiometer (MODIS) sensor onboard the Terra satellite; they found very good
agreement between the MODIS and SGLI cloud products for both water and ice clouds over both ocean and land
surfaces. Because cloud retrievals from MODIS and SGLI are based on the same retrieval framework of Nakajima
and King (1990) and similar types of cloud reflectance data, it is very important to assess the quality of SGLI cloud
products by using data of different natures obtained from different observation techniques, such as those obtained
from surface observations. Damiani et al. (2019) compared SGLI-observed COD with surface-observed values
obtained using different instruments, including a sky radiometer and a pyranometer; they found reasonably good
agreement between the SGLI and surface observations, but their study was limited to an observation period of a
few days (16 days) with very few samples for comparison and for water cloud COD only. By contrast, the present
study is designed to use long-term observation data from multiple sites to assess the quality of the properties of
both water and ice clouds.
This paper is organized as follows. Sections 2 and 3 describe the data and the study method, respectively.
Section 4 presents and discusses the results. Finally, section 5 summarizes the main findings of this study.

**2. Data**
**2.1. SKYNET**
Data from SKYNET sites in Japan (Table 1) for 2018–2020 were used in the present study. These sites have
different atmospheric backgrounds: Chiba and Sendai are urban sites, whereas Hedo-misaki, Fukue-jima, and
Miyako-jima are located on the coast of the east China Sea, where a different air mass—either marine or long-
range transported continental—prevails in different seasons (Khatri et al., 2010, 2014a), making them unique for
studying aerosols and clouds. Except for Sendai, all these sites are "super sites" of SKYNET, being equipped with
various instruments for observing aerosols, clouds, radiation, and meteorology. We used two types of SKYNET
data as described below.

**2.1.1. Sky radiometer**
Nakajima et al. (2020) described in detail the sky radiometer technology of SKYNET. Although sky radiometer
data have been used widely to study aerosol properties (e.g., Hashimoto et al., 2012; Wang et al., 2014; Khatri
et al., 2016; Mok et al., 2018; Irie et al., 2019), retrievals of ozone (Khatri et al., 2014b), water vapor (Campanelli
et al., 2014), and clouds (Khatri et al., 2019) are also possible from sky radiometer observations. The present study
used cloud properties retrieved from a sky radiometer (POM-02; PREDE Co., Ltd., Japan) that observed spectral
zenith radiances at 10-min intervals. Of 11 wavelengths between ultraviolet and near-infrared, the zenith radiances
observed at 0.87, 1.02, and 1.627 μm were used to obtain COD and CER via a cloud retrieval algorithm by Khatri
et al. (2019). Aerosol observations made at the wavelengths of 0.38, 0.4, 0.5, 0.675, 0.87, and 1.02 μm under clear

sky conditions were used to derive the temporal (monthly) variations of the calibration constants for the wavelengths of 1.627 μm (absorbing) and 0.87 μm and 1.02 μm (non-absorbing) to convert the observed signals into transmittances. These spectral transmittances were then combined with spectral surface reflectance and precipitable water content (PWC) to retrieve COD and CER simultaneously via an optimal method (Rodgers, 2000). The surface reflectance and PWC data were obtained from MODIS and Modern-Era Retrospective Analysis for Research and Applications, Version 2 (MERRA-2), respectively. In retrieving the properties of water clouds, single-scattering properties generated for spherical water cloud droplets estimated from Mie calculations were used, whereas such databases corresponding to the Voronoi model of irregular shapes for ice particles (Ishimoto et al., 2012) were used in retrieving the properties of ice clouds.

### 2.1.2. Pyranometer

Each SKYNET site in Japan is equipped with a pyranometer (Kipp and Zonen, Holland) to measure downwelling global irradiances. Because the global irradiance over Miyako-jima was observed for only a limited study period, the data from the remaining four sites were used in this study. The observed global irradiances were for the spectral range of 0.315–2.8 μm and for a temporal resolutions of 60 s for Sendai and 20 s for the other sites.

### 2.2. SGLI

We used the Level 2.0 (Version 2.0) cloud products of SGLI (Nakajima et al., 2019). The GCOM-C satellite carrying the SGLI sensor is timed to cross the equator from north to south at approximately 10:30 AM local time, and the spatial resolution of the SGLI cloud product is 1 km × 1 km at nadir. The SGLI cloud products were retrieved using the CAPCOM cloud property retrieval algorithm (Nakajima and Nakajima, 1995; Kawamoto et al., 2001; Nakajima et al., 2019), in which the 1.05- and 2.21-μm channels were used as the non-absorbing and absorbing wavelengths, respectively, while developing the look-up table (LUT) of cloud reflectance (Nakajima and King, 1990). The LUTs for water clouds and ice clouds were developed using the single-scattering properties of spherical water cloud droplets calculated using Mie theory and non-spherical Voronoi particles (Ishimoto et al., 2012). Along with those reflectance data, the algorithm also used ancillary data such as vertical profiles of temperature, water vapor, and surface reflectance while retrieving cloud properties (Nakajima et al., 2019).

### 3. Study method

Depending on the departure of the viewing angle of the satellite sensor from nadir, a parallax—that is, a shift in cloud position (longitude and latitude) from that corresponding to the surface—can occur, and correcting this parallax is important when comparing oblique-view satellite products with observations made either at nadir view from space (e.g., Khatri et al., 2018a) or at zenith view from the surface (e.g., Khatri et al., 2018b). Therefore, the SGLI cloud products were parallax-corrected by using information about cloud-top height, the zenith and azimuth angles of the satellite, and the position (latitude and longitude) of the observation site. Then, if all satellite pixels with 5 × 5 coverage and the observation site at the central pixel were cloudy, they were used to calculate the average values of COD and CER. They were then compared with the sky radiometer values observed at the surface

within ±30 min of the SGLI observation time. Such averaging practices can address cloud movement (Cess et al.,
1996) and are common in validating satellite cloud products using surface observation data. For example, Dong
et al. (2008) and Yan et al. (2015) compared CERES-MODIS cloud properties averaged over a 30 km × 30 km
square and a circle of 20-km radius around the observation site, respectively, with surface observation values
averaged over a 1-h period.
The pyranometer-observed global irradiances were further used to assess the quality of the SGLI cloud products.
For this purpose, the SGLI cloud properties and ancillary data such as PWC from MERRA-2 and spectral surface
reflectance from MODIS were used in an RSTAR radiative-transfer model (Nakajima and Tanaka, 1986, 1988) to
calculate downwelling global irradiances in the 0.315–2.8-µm spectral range. The single-scattering properties
obtained from Mie calculations for water clouds and those corresponding to the Voronoi model for ice clouds were
used for water and ice clouds, respectively. The modeled global irradiances of the 5 × 5 pixels centered on the
observation site were then averaged to compare with the values observed at the surface for ±5 min centered on the
SGLI observation time.
To quantify the degree of agreement between the SGLI and surface observations, the mean bias error (MBE),
root-mean-square error (RMSE), and correlation coefficient (*r*) values were calculated as
$MBE = \frac{1}{n}\sum_{i=1}^{n}(G_i - S_i),$       (1)
$RMSE = \sqrt{\sum_{i=1}^{n}\frac{(G_i - S_i)^2}{n}},$       (2)
$r = \frac{n(\sum_{i=1}^{n}G_i S_i) - \sum_{i=1}^{n}G_i \sum_{i=1}^{n}S_i}{\sqrt{[n\sum_{i}^{n}G_i^2 - (\sum_{i=1}^{n}G_i)^2][n\sum_{i}^{n}S_i^2 - (\sum_{i=1}^{n}S_i)^2]}},$       (3)
where $G_i$ and $S_i$ are surface and satellite observations, respectively, and *n* is the total sample count.

**4. Results and discussion**
**4.1. Comparison between SGLI-observed and sky radiometer-observed cloud properties**
**4.1.1. Overall comparison**
The COD values from the sky radiometer and SGLI are compared in Figure 1(a) and (b) for water clouds and
ice clouds, respectively. In general, the values from the two different sources agree reasonably well for both cloud
types. The *r* value for water clouds is higher than that for ice clouds, suggesting that the temporal variations of
COD from the sky radiometer and SGLI are more consistent with each other for water clouds than for ice clouds.
The MBE values are positive and nearly the same for water and ice clouds. Overall, these positive MBE values
suggest smaller COD from SGLI than from the sky radiometer for both water and ice clouds, but upon closer
inspection, Figure 1 indicates that whether the COD from SGLI is an overestimate or an underestimate depends
on the COD value; we have underestimated values from SGLI for relatively high COD for both water and ice
clouds, whereas most of the data samples show an overestimated COD from SGLI when they are less than ~20
and ~10 for water and ice clouds, respectively. A literature review also suggests similar results in the past for COD
observed by other remote-sensing tools. For example, King et al. (2013) and Liu et al. (2013) showed
overestimated (resp. underestimated) COD for values less (resp. greater) than ~20 when they compared MODIS

COD with values obtained from in situ aircraft observations and a multifilter rotating shadowband radiometer, respectively. Nakajima et al. (1991) also found overestimation (resp. underestimation) of COD for values less (resp. greater) than ~10 while comparing their products retrieved from cloud reflection measurements with those obtained from in situ aircraft observations. Khatri et al. (2018b) also found similar results when they compared COD values observed by MODIS and the Advanced Himawari Imager with surface-observed values. The consistency of Figure 1 with those previous studies indicates that reflectance-based COD from satellite retrieved by assuming PPH cloud layers (i.e., by using one-dimensional (1D) radiative transfer theory) can be underestimated (resp. overestimated) for optically thin (resp. thick) clouds irrespective of sensor type. It can be noted in a Nakajima-King diagram that COD increases (decreases) with the decrease (increase) of value corresponding to absorbing wavelength even without any change of value corresponding to non-absorbing wavelength. Since satellite-observed signal corresponding to absorbing wavelength is mostly from upper portions of clouds, it can be less than the value that can result from whole cloud layers. Under such condition, retrieved COD can be overestimated. However, subpixel inhomogeneity is commonly known to underestimate retrieved COD in satellite observation when clouds are assumed to be PPH layers (Cahalan et al., 1994). Cahalan et al. (1994) suggested that such subpixel inhomogeneity effect, which is also called as "plane-parallel albedo bias", is very weak for thin clouds and very thick clouds reaching albedo saturation, but strong for moderately thick clouds. Thus, these two different effects may counter each other to increase or decrease COD. The less influence of "plane-parallel albedo bias" for thin clouds may result SGLI-observed CODs higher than sky radiometer-observed values for relatively thin clouds. On the other hand, the opposite for relatively thick clouds could be the result of dominant effect of "plane-parallel albedo bias". A detailed investigation is required in the future to further clarify the mechanism for such results.

The CER values from the sky radiometer and SGLI are compared in Figure 2 (a) and (b) for water clouds and ice clouds, respectively. The CER values show poorer agreement than do the COD values in the comparisons for both water and ice clouds. There can be a number of reasons for such poorer agreement for CER. First, unlike surface-based sky radiometer, the upper portions of clouds are sampled more readily than lower parts in space-based SGLI. Since cloud-droplets can have vertical inhomogeneity with upper cloud portions containing both relatively large-sized (e.g., an adiabatic growth at the beginning of cloud generation) as well as small-sized (e.g., entrainment of dry air at the cloud top, collision-coalescence process) particles, CERs retrieved from SGLI observations can become both larger and smaller than those retrieved from sky radiometer observations, as noted in Figure 2, depending on vertical inhomogeneity of clouds. Further, as the absorbing wavelengths, which are critical for CER retrievals, corresponding to current SGLI and sky radiometer cloud retrieval algorithms are 2.2 and 1.6 μm, respectively, these different wavelengths can have different absorptions to further enhance the difference in CER between SGLI and sky radiometer. Except them, quality of data samples used for the comparison holds an important position to determine the comparison metrics, such as $r$ value, RMSE, and MBE. For example, if we screen data shown in Figures 1 and 2 by selecting only those that have coefficient of variation (COV), i.e., the ratio of standard deviation value to the mean value, less than 0.2 for CODs of both sky radiometer and SGLI, the comparison metrics, including those for CER comparisons, can have different values (Figure 3). CODs with

COV less than 0.2 for sky radiometer (SGLI) can represent data samples having very less temporal (spatial)
variations in sky radiometer (SGLI) observations, indicating relatively strict data screening criteria for comparison.
Figure 3(a) shows a very good agreement for CER comparison for water clouds. On the other hand, the comparison
metrics corresponding to CER comparison for ice clouds are still poor because a limited number of samples show
considerably large difference between sky radiometer and SGLI. However, on the other side, it is still encouraging
to see a considerable number of samples falling around 1:1 line in Figure 3(b). Nevertheless, Figure 3 suggests an
important role of data handling procedure while evaluating cloud properties obtained from space-based
observations with those from surface-based observations. Further, as the number of scattering within cloud layers
increases with the increase of cloud thickness, COD can be suggested to play an important role in retrieved CER
value. The influence of COD on retrieved CER in satellite remote sensing has been discussed in detail from both
theoretical (e.g., Nakajima and King, 1990) as well as observation perspectives (e.g., Zhang and Platnick, 2011).
Similarly, Khatri et al. (2019) showed the influence of COD on retrieved CER for surface-based sky radiometer.
Figure 4 shows the relationship between CER difference, i.e., $\Delta$CER ($CER_{SGLI}$-$CER_{skyrad}$) and $COD_{SGLI}$ for water
clouds and ice clouds. In general, Figure 4 suggests a negative correlation between $\Delta$CER and $COD_{SGLI}$. Such
negative correlation is relatively less prominent for ice clouds than for water clouds, which can probably due to
irregular shapes of ice cloud particles that adds complexity while retrieving cloud properties in both sky radiometer
and SGLI observations. Figure 4(a) suggests that SGLI and sky radiometer CERs, in general, may have relatively
close agreement for CODs around 20. Note that CODs from SGLI and sky radiometer also show relatively close
agreement for CODs around 20, as discussed above. Figure 4(a) further suggests that CER values from SGLI can
be higher (lower) than sky radiometer values when clouds are relatively thin (thick). This result again coincides
with relatively higher values of COD from SGLI than those from sky radiometer for relatively thin (thick) clouds.
On the other hand, Figure 4(b) suggests that relatively very large difference in CER between SGLI and sky
radiometer can generally occur for relatively thin clouds. Note that retrieved CERs can have larger uncertainties
for optically thinner clouds in both surface and satellite retrievals (Khatri et al., 2019; Nakajima and King, 1990).
Nevertheless, Figure 4 suggests that CER difference between SGLI and sky radiometer can vary differently
depending on COD value, suggesting COD as one important candidate for CER difference between them. Along
with these factors, differences in ancillary and surface reflectance data in the retrieval algorithms of SGLI and sky
radiometer may also contribute partially to bring differences in retrieved values of CER as well as COD between
SGLI and sky radiometer. Although such manifold factors can be responsible for differences in CER values
between SGLI and sky radiometer, most of the data samples show higher CER values from SGLI than from the
sky radiometer, resulting in negative values of MBE for both water and ice clouds. This result is in line with
previous studies that showed higher values from satellite observations compared with values obtained from surface
and/or aircraft observations (e.g., Painemal and Zuidema, 2011; Chiu et al., 2012; King et al., 2013).

The comparison results discussed above suggest some future research scopes. Since cloud-droplet vertical

inhomogeneity can have important effects on retrieved cloud properties for both space- and surface-observation
data, future studies may effectively implement observation data of active sensors, such as surface-observation
based lidar, as well to improve and strengthen the quality assessment of CER values obtained from SGLI and other

similar satellite sensors. Furthermore, CER retrievals from SGLI (sky radiometer) may be extended for absorbing wavelength of 1.6 μm (2.2 μm) for further improving and strengthening such quality assessment studies as well as expanding our understanding regarding CER property. In addition, along with sky radiometer, other surface-based radiometers, such as rotating shadow-band spectro-radiometer (Khatri et al., 2012; Takamura and Khatri, 2021), that have wide field of view (FOV) can be brought in use for remote sensing of cloud properties from surface and to validate space-observed cloud properties more rigorously.

### 4.1.2. Comparison by separate cloud type

The SGLI cloud product also contains information about cloud type, which is determined based on COD and cloud top pressure (CTP), similar to the cloud classification method of the International Satellite Cloud Climatology Project. The data for water clouds shown in Figures 1(a) and 2(a) correspond to the altostratus, nimbostratus, stratocumulus, and stratus cloud types. Figure 5 compares the sky radiometer and SGLI cloud properties for each cloud type. Since comparison between SGLI and sky radiometer is performed for spatial and temporal averages of SGLI and sky radiometer observations, respectively, the cloud type used in this study corresponds to the pixel located at the center of 5x5 SGLI pixels, which includes observation site. Of these four types of clouds, the first two are mid-level clouds and the last two are low-level clouds, which have CTP value of 440–680 hPa and greater than 680 hPa, respectively. Similarly, altostratus and stratocumulus have COD values of 3.6–23, but nimbostratus and stratus have COD values greater than 23. In Figure 5, stratus clouds show the best agreement; compared to the other types of clouds, stratus clouds are more uniform and flatter, thereby being the closest to PPH cloud layers. After stratus clouds, nimbostratus clouds suggest the next-best agreement, although some CER values show large deviations from the 1:1 line. Because nimbostratus is a thick mid-level cloud, ice crystals or their combination with liquid cloud droplets—including supercooled droplets—can form in and around the cloud top, although middle and lower cloud portions can contain water cloud droplets. Under such conditions, retrievals from SGLI by considering the cloud phase as being water can affect the retrieved products significantly, especially CER, which in turn can cause considerably large differences from the CER observed from the surface. To some extent, the results from the sky radiometer can also be affected. But, since the sky radiometer observes from the surface, the dominant fractions of water in the middle and lower parts of such clouds have important influences on surface-observed radiances, which may make considering the water cloud phase reasonably valid in retrieval of cloud properties from surface observations for such conditions. On the other hand, altostratus and stratocumulus clouds show moderate agreement for COD but poor agreement for CER. Because these clouds can have COD values ranging from 3.6 to 23, large differences in CER comparison can arise, especially for relatively thin clouds. This is because the uncertainties in CER retrievals can be larger for thin clouds than for thick clouds in both sky radiometer and SGLI retrievals. Additionally, the high-level altostratus cloud can comprise ice and/or supercooled droplets near the cloud top to affect SGLI retrievals, as discussed above. Regarding low-level stratocumulus clouds, they may not contain such ice and/or supercooled liquid particles, but their cloud tops can be quite inhomogeneous because they are generally clumps of thick and thin clouds, resulting in a higher degree of cloud heterogeneity, which in turn can have large effects on satellite retrievals, as revealed from both modeling

(e.g., Iwabuchi and Hayasaka, 2002) and observation (e.g., Várnai and Marshak, 2007). Because SGLI has a larger
FOV than does the sky radiometer, instrumental FOV could be the next important factor for the large difference
between the sky radiometer and SGLI results for such highly heterogenous stratocumulus clouds.

Figure 6 compares the sky radiometer-observed and SGLI-observed cloud properties for different types of ice

clouds. As shown, seven types of ice phase clouds were detected, of which cirrus, cirrostratus, and deep convective
are high-level clouds, altocumulus, altostratus, and nimbostratus are mid-level clouds, and stratocumulus is a low-
level cloud. Cirrus and altostratus clouds have COD values of less than 3.6. These thin clouds have values of both
COD and CER that deviate largely from the 1:1 line, suggesting that large differences between the sky radiometer
and SGLI results can occur for thin clouds. This is because retrievals become ambiguous, resulting in two possible
solutions in both satellite retrieval (Nakajima and King, 1990) and sky radiometer retrieval (Khatri et al., 2019)
for such thin clouds. Furthermore, the sky radiometer-observed values are averages of $\pm 30$ min centered on the
SGLI overpass time, making it possible to include some nearby thick clouds not included in the $5 \times 5$ pixels of
SGLI observation, given that the wind speed can be high at high altitudes. Cirrostratus followed by altostratus
occupy significant numbers of the ice cloud data. Furthermore, these cloud types agree better than do the other
types; they are generally uniform stratiform (layered) genus-type, that is, closer to PPH cloud layers than are the
other types of clouds. However, despite having the best agreement, some considerably large differences between
the sky radiometer and SGLI results still exist; these could be due to high wind speed, especially for cirrostratus,
a mixture of both water and ice cloud droplets, especially for altostratus, and the irregular shapes of ice crystals.
Deep convective and nimbostratus clouds have COD values of greater than 23. Although the top layers of these
clouds generally contain irregularly shaped ice crystals, their middle and lower parts can contain water cloud
droplets and/or supercooled droplets, making it difficult to retrieve cloud properties from both SGLI and the sky
radiometer by using a database of a single type of cloud phase. These thick clouds suggest fairly good agreement
between the sky radiometer and SGLI cloud properties, as do the low-level stratocumulus clouds detected as ice
clouds by SGLI. Note that there appears a data sample with mean COD for SGLI less than 23 in Fig. 6(c). Though
ISCCP defines deep convective cloud with COD greater than 23, the anvil portion of deep convective clouds can
have COD less than 23. Thus, a part of cloud pixels around the central pixel is likely to be anvil cloud to result
mean COD less than 23 for that case. Overall, the above comparison results for different types of clouds for both
water and ice phases reveal that cloud properties retrieved from the sky radiometer and SGLI can agree better if
the clouds are relatively uniform, flat, and thick.

**4.1.3. Effects of solar and satellite viewing geometries on comparison results**

Satellite cloud products retrieved by assuming PPH cloud layers can have biases depending on solar zenith

angle (SZA) (Kato and Marshak, 2009) and satellite viewing zenith angle (VZA) (Várnai and Marshak, 2007). To
understand how such SZA and VZA biases might influence the differences between SGLI and sky radiometer
cloud properties, comparisons are performed by separating the data for each SZA and VZA greater than and less
than 30°. The comparison results corresponding to water and ice clouds are shown in Figures 7 and 8, respectively.
Note that the SZA and VZA values used in this study correspond to the pixel located at the center of 5 x 5 SGLI
pixels. To understand further how SZA and VZA biases might influence the comparison results, we calculated the
mean values of SZA and VZA for different levels of agreement in the sky radiometer and SGLI comparisons. In
other words, these mean values were calculated by identifying very good agreement (difference less than 30%),
moderate agreement (difference within 30%–60%), poor agreement (difference within 60%–90%), and very poor
agreement (difference greater than 90%), where the difference is $|x_{SGLI} - x_{sky}|/x_{sky} \times 100\%$ and $x$ represents
COD or CER. The mean values are summarized in Tables 2 and 3 for water and ice clouds, respectively. Figures 7
and 8 both show better agreements between the sky radiometer and SGLI COD values for SZA < 30° than for SZA
> 30°. Also, Table 2 suggests that increasing SZA increases the COD difference for water clouds. Although not
distinct as in the case of water clouds, the COD difference for ice clouds also indicates its dependency on SZA in
Table 3. These results suggest possible SZA bias in SGLI-observed COD and so its influence on the COD
differences between the sky radiometer and SGLI. Both observations (e.g., Loeb and Davies, 1997) and radiative-
transfer model simulations (e.g., Kato et al., 2006) suggest that COD retrieved by assuming PPH cloud layers
increases with SZA because the horizontal leakage of radiation from cloud sides decreases relative to overhead
Sun (Fu et al., 2000) and cloud sides have a greater opportunity to intercept more solar radiation for oblique Sun
to increase the cloud-top-leaving radiance (Loeb et al., 1997). On the other hand, there appears to be no clear
improvement in COD comparison between the sky radiometer and SGLI with increasing or decreasing VZA.
However, as revealed from Figure 7, the COD comparison between the sky radiometer and SGLI for water clouds
may deteriorate considerably when both SZA and VZA become large. Supporting this result, Table 2 shows higher
values of VZA for cases of moderate and poor agreement than for very good and moderate agreement for water
clouds. However, there seems to be no clear signature of the dependence of COD difference on VZA for ice clouds
in Table 3. Liang and Di Girolamo (2013) suggested that satellite COD retrieved under the assumption of PPH
cloud layers can either decrease or increase with VZA depending on the competition among multiple factors
governed by SZA, relative azimuth angle (RAZ), and cloud inhomogeneity. This can plausibly explain the unclear
effects of VZA on the COD comparisons shown in Figures 7 and 8 and summarized in Tables 2 and 3. For CER,
it is almost impossible to say how SZA and VZA influence the CER comparisons shown in Figures 7 and 8,
although to some extent it is likely that water clouds exhibit better agreement for low SZA than for high SZA.
Coinciding with the results shown in Figures 7 and 8, Tables 2 and 3 also discard the influences of SZA and VZA
on the CER differences between the sky radiometer and SGLI. Because the cloud properties observed from SGLI
and the sky radiometer depend strongly on cloud type, as discussed above, and CER retrievals have larger
uncertainties than do COD retrievals, these factors possibly diluted the influences of SZA and VZA on the CER
differences between the sky radiometer and SGLI.

SZA and VZA biases on retrieved cloud properties for other satellite sensors—including MODIS—have been

studied widely, but such studies for the SGLI sensor have been lacking to date. We further analyzed the SGLI
observed reflectance ($R$) to shed further light on possible biases on the SGLI-retrieved cloud properties. For this
purpose, the SGLI-observed $R$ (1.05 µm) data with values of less than 1 for 500 pixels centered on the Chiba
observation site were analyzed. Those data correspond to 2020. $R$ values corresponding to different values of COD
(COD = 2±1, 4±1, 8±1, 16±1, 32±1, 64±1) and SZA (SZA = 20°±1°–60°±1° in 5° intervals) were binned by
accounting for the relative azimuth angle (RAZ), that is, the difference in azimuth angles between Sun and satellite.
Figure 9 shows the $R$-VZA relationships for these values of COD and SZA. Negative (resp. positive) VZA
corresponds to RAZ greater (resp. less) than 90°, representing forward (resp. backward) scattering. Because the
$R$-VZA relationship is similar for ice clouds, it is not shown here. The data fluctuations in Figure 9 for the same
values of COD and SZA suggest variations of CER and surface and atmospheric conditions. To tally such observed
$R$-VZA relationships, we again calculated $R$ (1.05 µm) for the COD values of 2, 4, 8, 16, 32, and 64 with a fixed
CER of 8 µm and SZA values of 20°–60° with intervals of 5° by assuming PPH cloud layers (Figure 10). These
calculations were performed for RAZ values of 135° and 45° to understand the characteristics of forward and
backward scattering, respectively. The calculated results shown in Figure 10 reveal that the $R$-VZA relationship
for ideal PPH cloud layers can have different shapes depending on SZA. For low SZA, the differences in $R$ between
the forward and backward scattering directions are relatively small, increasing gradually with increasing SZA; for
high values of SZA, the $R$ values in the forward scattering directions are higher than those in the backward
scattering direction. The shapes of the $R$-VZA relationship for ideal PPH cloud layers, which correspond to the
LUT databases of satellite retrieval algorithms, are different than those shown in Figure 9 corresponding to actual
observations, suggesting that three-dimensional (3D) cloud effects on observed signals are not captured well in
1D radiative-transfer calculations. Liang and Girolamo (2013) suggested that the observed VZA dependence of
COD (or $R$) is the weighted sum of different competing factors associated with Sun and satellite positions and
cloud inhomogeneity. For example, Várnai and Marsak (2007) found increased values of COD with increasing
VZA in both the forward and backward scattering directions, and they suggested that the dark gaps between cloud
fields could be filled up by brighter cloud sides through photon leakage when partly cloudy scenes are viewed
more obliquely, leading to higher values of COD in both the forward and backward scattering directions. On the
other hand, Loeb and Coakley (1998) found decreased and increased COD in the forward and backward scattering
directions, respectively, which they attributed to shadowing and illumination. Nevertheless, the most important
information revealed from Figures 9 and 10 is that 1D radiative-transfer theory may hardly capture the features of
$R$ observed by SGLI, suggesting SZA and VZA biases on the retrieved cloud properties from SGLI, similar to
other satellite sensors including MODIS.

**4.2. Comparison between modeled and observed global irradiances**
Because surface-observed global irradiances vary strongly with cloud variation (Khatri and Takamura, 2009;
Damiani et al., 2018), they can also help to justify the comparison results discussed above. Specifically, surface-
observed global irradiances can be effective for evaluating satellite-observed COD values because the variation of
COD is more dominant than the variation of CER in the variation of global irradiance (Khatri et al., 2018b).
Figure 11 compares the measured and modeled global irradiances at the four observation sites that have
observation data for the whole study period for water clouds (Figure 11(a)) and ice clouds (Figure 11(b)). These
comparisons are for the mean values of the $5 \times 5$ SGLI pixels and ±5 min of surface observations centered on the
SGLI observation time. The measured global irradiances agree very well with the observed values for both water
clouds and ice clouds. In both cases, the correlations are very strong with values greater than 0.85. The RMSE

value for water clouds is smaller than that for ice clouds, suggesting that water cloud optical properties can reproduce global irradiance better than can ice cloud properties. Although not distinctly different, the absolute MBE value is smaller for water clouds than for ice clouds. Overall, these results suggest that retrieval accuracies are better for water cloud properties than for ice cloud properties in SGLI. Note that the COD values from the sky radiometer also agree better for water clouds than for ice clouds. The MBE values are negative for both water and ice clouds, suggesting that the modeled global irradiances are generally higher than the observed values. Such negative MBE values (overestimation of modeled global irradiances) can result from underestimated COD values in SGLI. This result again coincides with the positive MBE values for the comparisons of COD between the sky radiometer and SGLI shown in Figure 1. Furthermore, Figure 12 shows scatter plots for the normalized differences between the modeled and measured values and the observed values. The observed global irradiance can suggest the COD, with a low global irradiance corresponding to a high COD and vice versa. Figure 12 suggests overestimated values of the modeled global irradiance when the observed values are relatively low, suggesting underestimated COD in SGLI when the clouds are optically thick. This result again coincides with the comparison of COD between the sky radiometer and SGLI shown in Figure 1 and discussed in section 4.1.1. The fewer data samples for water clouds hardly suggest underestimated modeled irradiance (overestimated SGLI COD) when the observed global irradiance is relatively high, but it is somewhat evident in ice clouds. This result again supports the underestimation (resp. overestimation) of COD from SGLI when the clouds are relatively thick (resp. thin), as discussed in section 4.1.1.

Unlike the COD comparisons shown for different cloud types and SZA and VZA values, the measured and modeled global irradiances do not show distinct differences depending on cloud type and SZA and VZA values. This is likely due to the fact that the global-irradiance-observing pyranometer has a wide FOV. Khatri et al. (2019) discussed the importance of an instrument's FOV for cloud remote sensing.

**5. Conclusions**

The main findings of this study are summarized below.

1.  The COD values from SGLI agreed reasonably well with the values observed from the surface using a sky radiometer by showing correlation coefficient ($r$) values of ~0.8 and ~0.6, RMSE values of ~10 and ~8, and MBE values of ~3 and ~3 for water and ice clouds, respectively. There appears to be a tendency of underestimating (resp. overestimating) the COD in SGLI for relatively thick (resp. thin) clouds. By contrast, the CER comparisons showed poorer agreements than the COD values, with $r$ values of ~0.1 and ~0.3, RMSE values of ~7 and ~18 µm, and MBE values of ~−0.5 and ~−10 µm for water and ice clouds, respectively.

2.  Comparison analyses performed by separating cloud types revealed that relatively thin, possibly mixed, and horizontally inhomogeneous cloud types generally have larger discrepancies than do relatively uniform and flat types of clouds for both the water and ice phases of clouds.

3.  The COD differences between SGLI and the sky radiometer showed strong and weak dependencies on SZA for water and ice clouds, respectively, by showing increasing difference with increasing SZA. On the other

hand, only the COD difference for water clouds showed a weak dependency on VZA, with increased difference for high VZA.

4. Analyses of the SGLI-observed reflection as functions of SZA, VZA, and COD and similar values of computed reflection functions by using a 1D radiative-transfer model (assuming PPH cloud layers) were inconsistent with each other, indicating that the 1D model was insufficient for capturing 3D cloud effects on the observed signals and thereby SZA and VZA biases on the retrieved cloud properties.

5. The surface global irradiances calculated using the SGLI-observed cloud properties agreed very well with the surfaced-observed values, with $r$ values of ~0.9 and ~0.9, RMSE values of ~66 and ~91 $Wm^{-2}$, and MBE values of ~$-32$ and ~$-33$ $Wm^{-2}$ for water and ice clouds, respectively. These results further revealed higher values of modeled irradiances than observed values when the latter were relatively low, and vice versa. These results further justified the evaluations of SGLI COD performed using the sky radiometer by emphasizing that (i) the SGLI COD can be underestimated on average, (ii) water cloud properties may have better retrieval accuracies than do ice cloud properties, and (iii) the SGLI COD can be underestimated (resp. overestimated) for optically thick (resp. thin) clouds.

*Code availability.* Codes for data analyses are available from the corresponding author upon request.

*Data availability.* SGLI data can be downloaded from Global Portal System (G-portal) of JAXA (https://gportal.jaxa.jp/gpr/). Similarly, global irradiance data can be downloaded from SKYNET webpage (http://atmos3.cr.chiba-u.jp/skynet/). Cloud properties retrieved from sky radiometer observations can be obtained from the corresponding author upon request.

*Author contributions.* PK and TH developed the study framework. HI and TT generated data. PK, HL, TN, and HI developed algorithms.

*Competing interests.* The authors declare that they have no conflict of interest.

*Special issue statement.* This article is part of the special issue "SKYNET – the international network for aerosol, clouds, and so- lar radiation studies and their applications (AMT/ACP inter-journal SI)". It is not associated with a conference.

*Acknowledgements:* This research is supported by the 2nd Research Announcement on the Earth Observations of the Japan Aerospace Exploration Agency (JAXA) (PI No. ER2GCF211, Contract No. 19RT000370), a Grant-in-Aid for Scientific Research (C) 17K05650 from Japan Society for the Promotion of Science (JSPS), and "Virtual Laboratory for Diagnosing the Earth's Climate System" program of MEXT, Japan.

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

Table 1. SKYNET sites for surface observation data

| Location | Longitude (°E) | Latitude (°N) |
|---|---|---|
| Chiba | 140.104 | 35.625 |
| Hedo-misaki | 128.248 | 26.867 |
| Fukue-jima | 128.682 | 32.752 |
| Miyako-jima* | 125.327 | 24.737 |
| Sendai | 140.839 | 38.259 |

*Missing of surface radiative flux data

Table 2. Mean values of SZA and VZA for different levels of differences in water cloud properties observed by
sky radiometer and SGLI

| Diff. Range | COD | | | CER | | |
|---|---|---|---|---|---|---|
| | SZA(°) | VZA(°) | N | SZA(°) | VZA(°) | N |
| 0-30% | 35.76±11.42 | 25.66±15.20 | 43 | 37.35±12.59 | 27.07±14.07 | 34 |
| 30-60% | 38.41±8.44 | 23.68±16.83 | 13 | 36.99±10.11 | 26.90±16.72 | 20 |
| 60-90% | 48.52±9.37 | 29.77±12.53 | 5 | 42.70±10.38 | 25.32±11.26 | 4 |
| > 90% | 53.10±2.84 | 29.05±5.28 | 3 | 43.03±7.16 | 14.58±12.80 | 6 |



Table 3. Mean values of SZA and VZA for different levels of differences in ice cloud properties observed by sky
radiometer and SGLI

| Diff. Range | COD | | | CER | | |
|---|---|---|---|---|---|---|
| | SZA(°) | VZA(°) | N | SZA(°) | VZA(°) | N |
| 0-30% | 33.62±12.67 | 26.96±12.67 | 63 | 31.29±12.51 | 25.97±14.27 | 53 |
| 30-60% | 32.09±12.65 | 27.60±14.46 | 72 | 33.62±13.27 | 27.63±12.32 | 30 |
| 60-90% | 33.75±12.22 | 25.89±13.52 | 29 | 32.95±13.98 | 27.31±12.19 | 25 |
| > 90% | 44.20±12.25 | 26.08±13.96 | 9 | 36.62±12.34 | 27.85±14.02 | 52 |




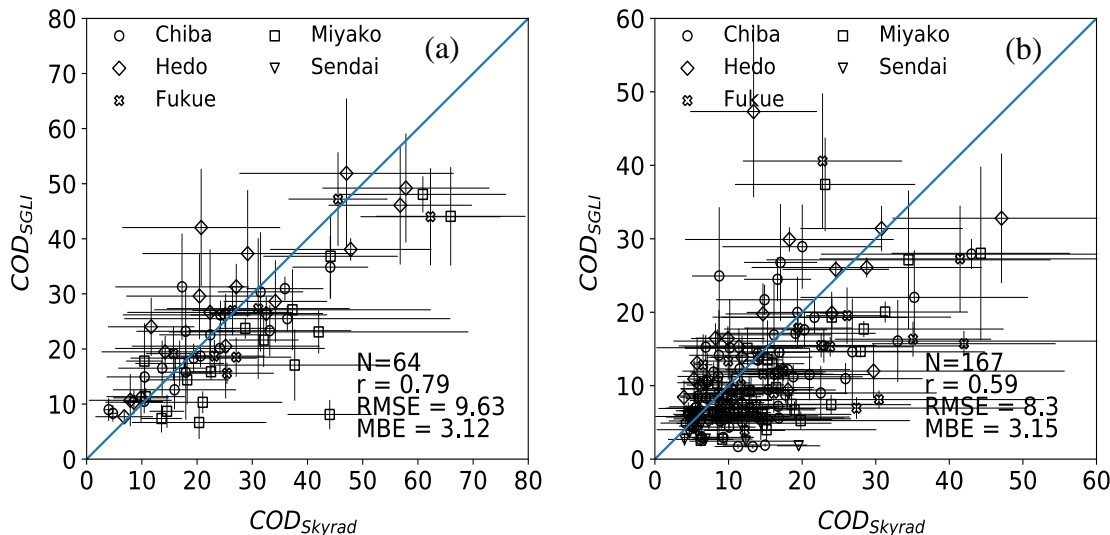

**Figure 1. Comparison of COD between sky radiometer and SGLI for (a) water clouds and (b) ice clouds**
**for data collected over SKYNET sites.**

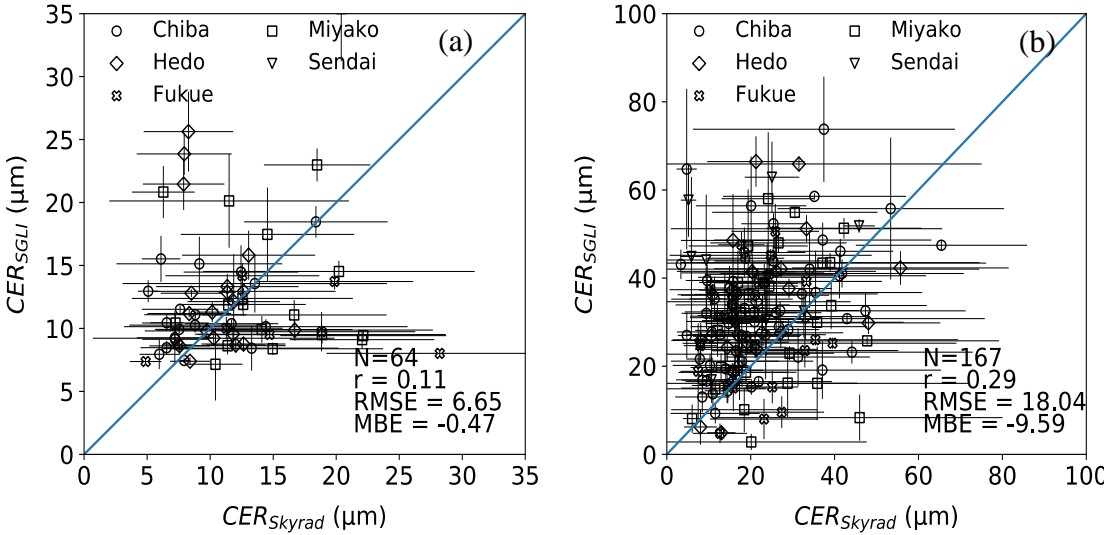

**Figure 2. As Figure 1 but for CER.**


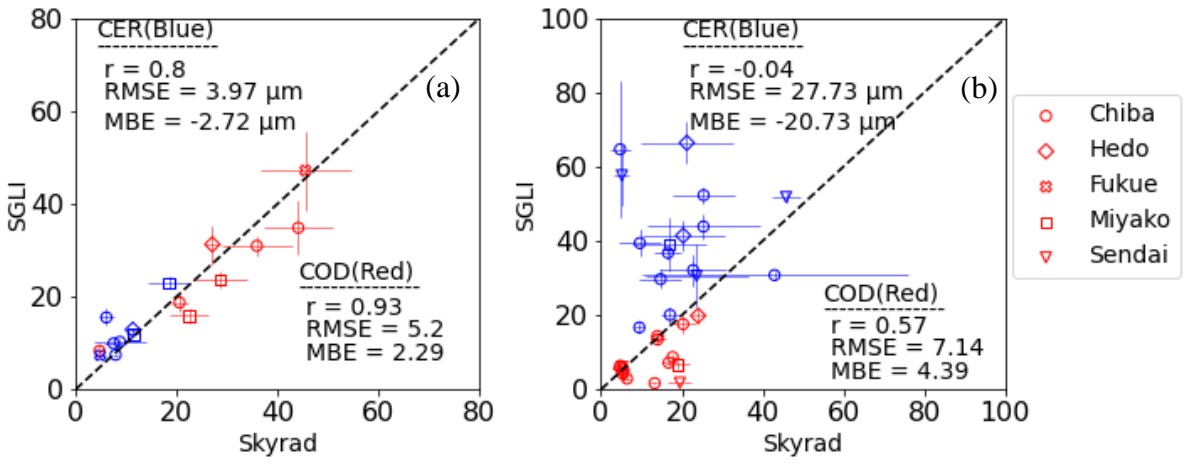

Figure 3. Comparison of cloud properties (COD and CER) between sky radiometer and SGLI for (a) water

clouds and (b) ice clouds by selecting data samples with coefficient of variation (COV) less than 0.2.


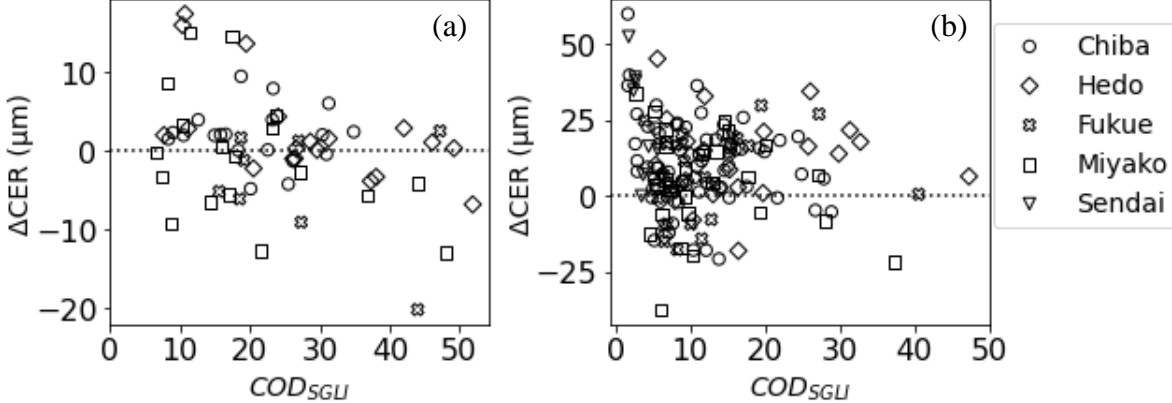


Figure 4. Comparison between △CER (CER$_{SGLI}$-CER$_{skyrad}$) and SGLI$_{COD}$ for (a) water clouds and (b) ice

clouds.





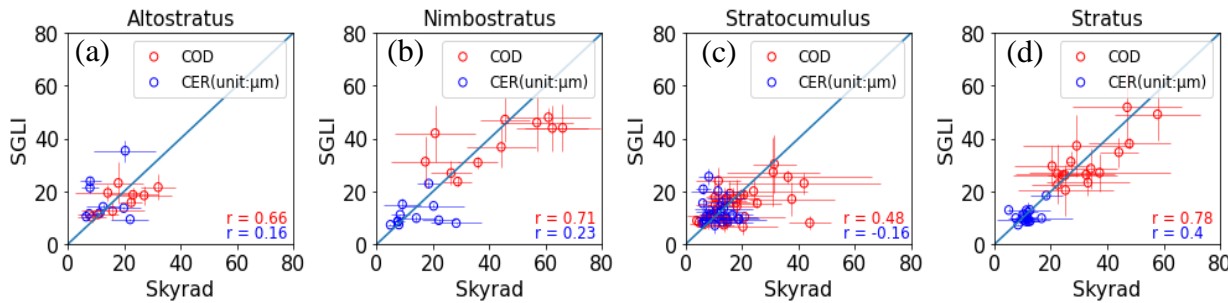


**Figure 5. Comparison between sky radiometer-observed and SGLI-observed water cloud properties for different types of clouds. The cloud type corresponds to the central pixel of 5x5 SGLI pixels.**


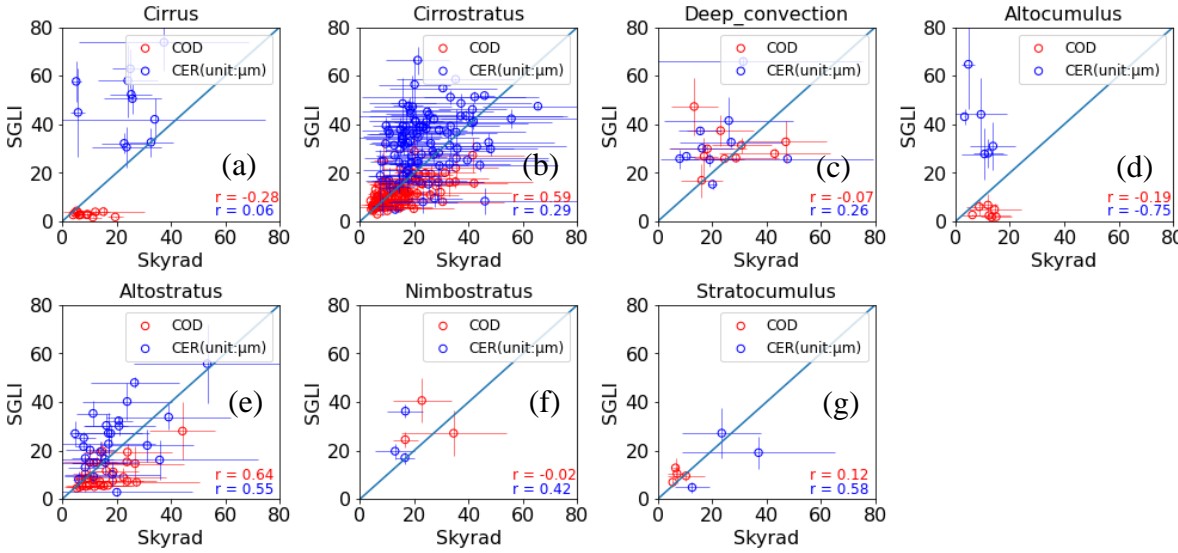


**Figure 6. As Figure 3 but for ice cloud properties.**

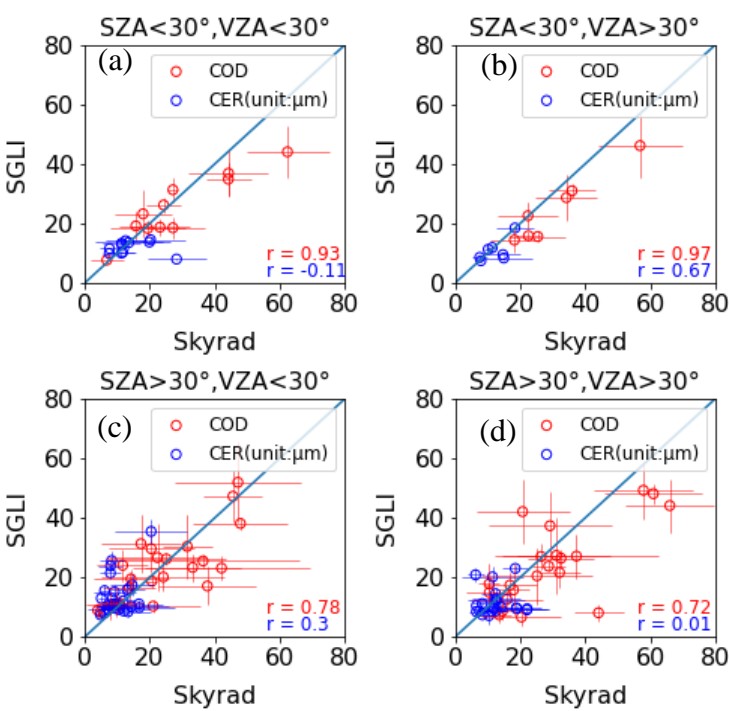


**Figure 7. Comparison between sky radiometer-observed and SGLI-observed water cloud properties for each SZA and VZA greater than and less than 30°. The SZA and VZA values correspond to the central pixel of 5x5 SGLI pixels.**



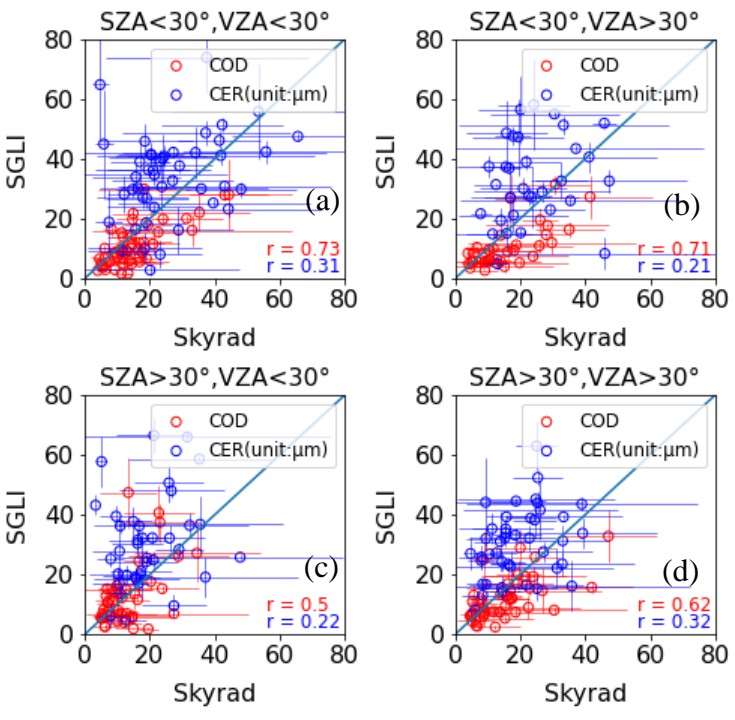


**Figure 8. As Figure 5 but for ice cloud properties.**

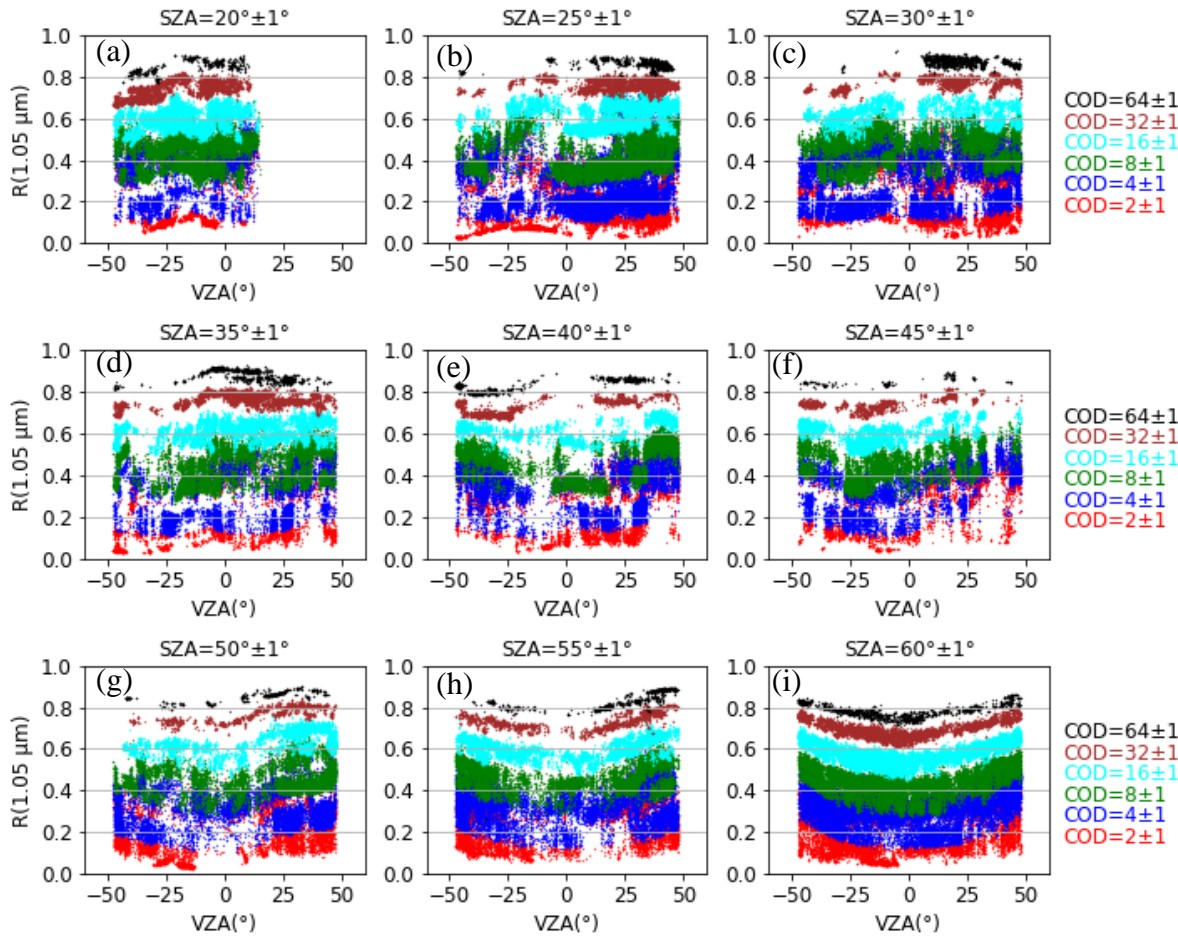


**Figure 9. Scatter plots of observed *R* (1.05 μm)–VZA relationships for different COD values of water clouds**
**at different SZA values. The data are for 500 pixels centered on the Chiba observation site in 2020. The**
**negative and positive VZA values represent the forward (RAZ > 90°) and backward (RAZ < 90°) scattering**
**directions, respectively.**








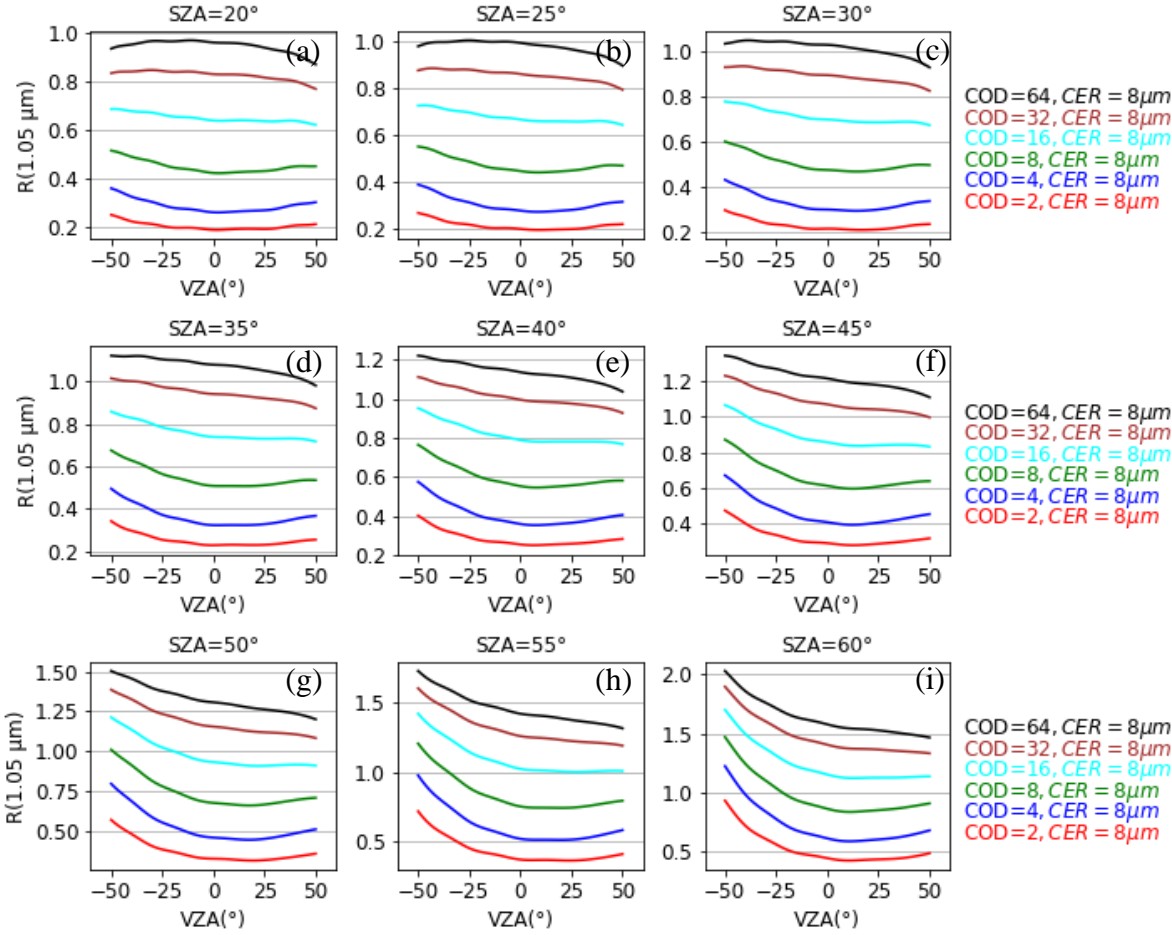


**Figure 10. Relationship between** $R$ **(1.05 μm) and VZA for different COD values and fixed CER of 8 μm for water clouds and different SZA values for assumption of plane-parallel cloud layers. The negative and positive VZA values correspond to RAZ values of 135° and 45° representing the forward and backward scattering directions, respectively.**









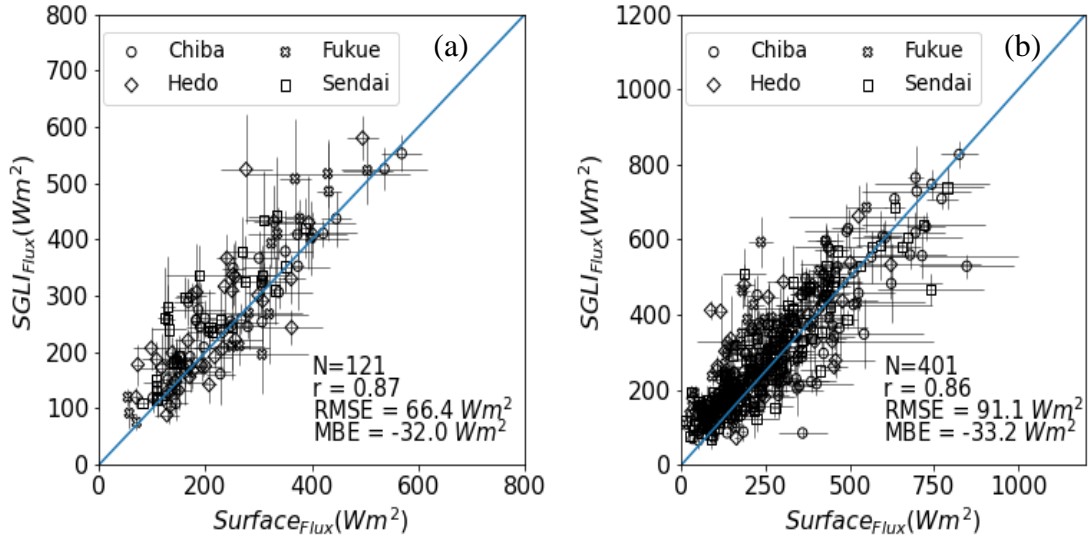


**Figure 11. Comparison of surface-observed global irradiances with values modeled using SGLI-observed cloud properties for (a) water clouds and (b) ice clouds.**


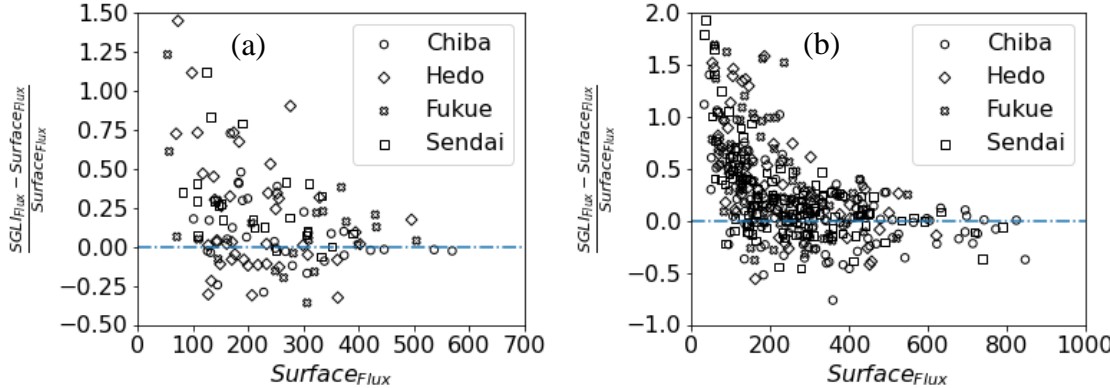


**Figure 12. Scatter plots for modeled and observed global-irradiance difference and observed global irradiance for   (a) water clouds and (b) ice clouds.**