# Peer review of "Quality assessment of Second-generation Global Imager"

_Atmospheric Measurement Techniques, 2021_

## Author Comment (AC1)

**Replies to comments of Reviewer 1**

Authors would like to express sincere thanks to an anonymous reviewer for his/her valuable comments and suggestions. We carefully revised the manuscript following the given suggestions and comments. Our replies to the comments and suggestions are given below.

It is important to assess SGLI-observed cloud properties using surface data, such as SKYNET data. The authors discuss the quality of the two most fundamental cloud properties—COD and CER of both water and ice clouds From SGLI. In fact, it is not easy to do this kind of assessment. In particular, the author should pay attention to the following issuesï¼š

--> Thank you very much for giving us positive feedbacks.

1. Objects ï¼ˆcloudsï¼‰ seen from satellites and the ground need to be substantially identical.

--> We agree with the reviewer. We made enough efforts to address this important issue in our study. First of all, we performed parallax-correction to all space-based SGLI cloud products before comparing them with surface-based sky radiometer results (**Page 4, Lines 136-141**). Though cloud systems move, however, in a statistical context, temporally averaging the surface measurements can be regarded to be equivalent to spatially averaging them over the satellite grid (Cess et al., 1995, 1996). Therefore, space-observed cloud properties are being evaluated using surface-observed values by taking spatial and temporal averages of space- and surface-observations, respectively (e.g., Dong et al., 2008; Nakajima et al., 2005; Takamura et al., 2009; Yan et al., 2015). This study follows similar procedure practiced over last few decades (**Pages 4-5, Lines 141-148**).

2. The algorithm that distinguishes between ice and water clouds requires precision and rigor to achieve good results.

--> We agree with the reviewer. As the focus of this study is to evaluate two most fundamental cloud properties—cloud optical depth (COD) and cloud-particle effective radius (CER)— of SGLI, for the purpose of this study, we retrieved cloud properties from sky radiometer by using scattering database for water (ice) cloud, if cloud detected by SGLI is water (ice). In other words, cloud phases for sky radiometer data analyses are consistent with SGLI observations to

better evaluate the qualities of SGLI cloud products (COD and CER) available for public use. Since retrieval algorithms for SGLI are being upgraded periodically, cloud phase detection algorithm may be upgraded in the future, and consequently cloud products may be upgraded as well.

3. As the authors say, the SGLI-observed CER exhibits poorer agreement than does the COD, with the SGLI values being generally higher than the sky radiometer values. And what's the reason? and how to improve? should be discussed in deatail.

--> As suggested by the reviewer, we discussed in detail regarding poor agreement of CER between sky radiometer and SGLI. We further discussed about future prospects of research for the improvement of CER retrievals and quality assessment studies. They are described as below in the revised manuscript (**Page 6, Line 222 - Page 8, Line 256**) .

[revised manuscript text omitted]

--> Thank you very much for pointing out typo. As suggested by the reviewer, there are inconsistent statements in the abstract and conclusion sections. The typo of conclusion is corrected in the revised manuscript (**Page 13, Line 447**).

---

## Author Comment (AC2)

**Replies to comments of Reviewer 2**

Authors would like to express sincere thanks to an anonymous reviewer for his/her valuable comments and suggestions. We carefully revised the manuscript by addressing given comments and suggestions. Our replies to the comments and suggestions are given below.

This paper validates the quality of SGLI cloud products through comparing with ground-based sky radiometer measurements for a few years. The authors have experiences for this sort of comparing work, thus their method seems adequate and contains enough content for publication. I, however, notice the following points that need to be clarified.

 -->Thank you very much for giving us positive feedbacks.

L174-176

'we have underestimated values from SGLI for relatively high COD for both water and ice clouds, whereas most of the data samples show an overestimated COD from SGLI when they are less than ~20 and ~10 for water and ice clouds, respectively.' Could the authors explain the mechanism to cause this tendency?

-->The overestimation (underestimation) of COD for relatively thin (thick) clouds in reflection-based remote sensing, similar to this study, has been reported in a number of past studies since a long ago (e.g., Nakajima et al., 1991; King et al., 2013; Liu et al., 2013; Li et al., 2014), though there lacks enough explanation for such observation-based results. As part of understanding the mechanism for such observation-based results, we generated a Nakajima-King plot by calculating radiances at the wavelengths of 1.05 and 2.21 μm, representing non-absorbing and absorbing wavelengths, respectively, for predefined values of COD and CER. The calculations are performed for water cloud, midlatitude summer atmospheric model, and black underlying surface (no surface reflectance) by assuming solar zenith angle, azimuth angle, and relative azimuth angle as 30°, 30° and 30°, respectively. Figure S1 shows a Nakajima-King plot for our calculated values. In Figure S1, we randomly chose two values of radiances at 2.21 μm, shown by red and blue lines, to understand how COD can change without any change of radiance at 1.05 μm. We further chose two values of radiances for 1.05 μm, shown by dashed lines. The crossing points of vertical and horizontal lines are indicated by black circles. Taking into account the positions of these black circles and curves corresponding to different values of COD and CER, it can be suggested from Figure S1 that COD increases (decreases) even

[Figure]

**Figure S1. Nakajima-King plot for calculated radiances at 1.05 and 2.21 μm for water cloud phase for solar zenith angle, satellite zenith angle, and relative azimuth angle of 30°, 30° and 30°, respectively. Radiances are calculated for midlatitude summer atmospheric model and black surface.**

without any change of observation signal at 1.05 μm, if the observation signal at 2.21 μm is underestimated (overestimated) due to any reason. Thus, the observation signal corresponding to absorbing wavelength is important not only for CER retrieval, but also for COD retrieval. It is important to note that satellite-observed signals at near-infrared wavelength, such as 2.21 μm used by SGLI, come mostly from upper portions of clouds. In other words, satellite-observed radiance at near-infrared wavelength can be less than the value that can result from whole cloud layers. Under such condition, retrieved COD can be overestimated. At the same time, it is important to note that subpixel inhomogeneity can underestimate satellite observation-based CODs when retrievals are performed by assuming clouds as plane-parallel horizontal layers (Cahalan et al., 1994). Such effect, which is also known as "plane-parallel albedo bias", is weak for thin clouds and very thick clouds that reach albedo saturation, but strong for intermediate values of COD (Cahalan et al., 1994). Thus, these two different effects may counter each other to increase or decrease COD. The less influence of "plane-parallel albedo bias" for thin clouds may result SGLI-observed CODs higher than sky radiometer-observed values for relatively thin clouds. On the other hand, the opposite for relatively thick

clouds could be the result of dominant effect of "plane-parallel homogenous bias". A detailed investigation is required in the future to further clarify the mechanism for such results.

Taking this valuable comment into account, we provide additional description, as below, in the revised manuscript (**Page6, Lines 186-199**).

*It can be noted in a Nakajima-King diagram that COD increases (decreases) with the decrease (increase) of value corresponding to absorbing wavelength even without any change of value corresponding to non-absorbing wavelength. Since satellite-observed signal corresponding to absorbing wavelength is mostly from upper portions of clouds, it can be less than the value that can result from whole cloud layers. Under such condition, retrieved COD can be overestimated. However, subpixel inhomogeneity is commonly known to underestimate retrieved COD in satellite observation when clouds are assumed to be PPH layers (Cahalan et al., 1994). Cahalan et al. (1994) suggested that such subpixel inhomogeneity effect, which is also called as "plane-parallel albedo bias", is very weak for thin clouds and very thick clouds reaching albedo saturation, but strong for moderately thick clouds. Thus, these two different effects may counter each other to increase or decrease COD. The less influence of "plane-parallel albedo bias" for thin clouds may result SGLI-observed CODs higher than sky radiometer-observed values for relatively thin clouds. On the other hand, the opposite for relatively thick clouds could be the result of dominant effect of "plane-parallel albedo bias". A detailed investigation is required in the future to further clarify the mechanism for such results.*

L189-190

'However, note that CER depends on not only the reflectance function of the absorbing wavelength but also the COD (Zhang and Platnick, 2011).' If so, how about showing CER comparison like Fig. 2 for thinner and thicker cloud cases separately?

--> Thank you for an important comment. First of all, we have made the above-mentioned statement very clear by adding some more sentences as below (**Page 7, Lines 221-225**).

*As the number of scattering within cloud layers increases with the increase of cloud thickness, COD can be suggested to play an important role in retrieved CER value. The influence of COD on retrieved CER in satellite remote sensing has been discussed in detail*

*from both theoretical (e.g., Nakajima and King, 1990) as well as observation perspectives (e.g., Zhang and Platnick, 2011). Similarly, Khatri et al. (2019) showed the influence of COD on retrieved CER for surface-based sky radiometer.*

As suggested by the reviewer, we further discussed about the influence of COD on CER difference between SGLI and sky radiometer. However, instead of showing a scatter plot diagram similar to Figure 2, we show the relationship between CER difference and COD. We think that the relationship between CER difference and COD can help to understand the dependence of CER difference on COD value more clearly than the comparison of CER values between SGLI and sky radiometer.

We provide additional description, as below, in the revised manuscript ( **Page 7, Lines 225-239**).

[Figure]

**Figure S2. Comparison between ΔCER (CER_SGLI-CER_skyrad) and SGLI_COD for (a) water clouds and (b) ice clouds.**

*Figure S2 (Figure 4 in the revised manuscript) shows the relationship between CER difference, i.e., ΔCER (CER_SGLI-CER_skyrad) and COD_SGLI for water clouds and ice clouds. In general, Figure S2 suggests a negative correlation between ΔCER and COD_SGLI. Such negative correlation is relatively less prominent for ice clouds than for water clouds, which can probably due to irregular shapes of ice cloud particles that adds complexity while retrieving cloud properties in both sky radiometer and SGLI observations. Figure S2(a) suggests that SGLI and sky radiometer CERs, in general, may have relatively close agreement for CODs around 20. Note that CODs from SGLI and sky radiometer also show relatively close agreement for CODs around 20, as discussed in section 4.1.1. Figure S2(a) further suggests*

*that CER values from SGLI can be higher (lower) than sky radiometer values when clouds are relatively thin (thick). This result again coincides with relatively higher values of COD from SGLI than those from sky radiometer for relatively thin (thick) clouds. On the other hand, Figure S1(b) suggests that relatively large difference in CER values between SGLI and sky radiometer can generally occur for relatively thin clouds. Note that retrieved CERs can have larger uncertainties for optically thinner clouds in both surface and satellite retrievals (Khatri et al., 2019; Nakajima and King, 1990). Nonetheless, Figure S2 suggests the CER difference between SGLI and sky radiometer can vary differently depending on COD value, suggesting COD as one important candidate for CER difference between them.*

L218-219

'the larger fractions of water in the middle and lower cloud portions may make considering the water cloud phase reasonably valid for surface observations.' Due to unreadability, could the authors rephrase the sentence?

--> As suggested by the reviewer, we have rephrased the sentence as below (**Page 8, Lines 276-279**).

*But, since the sky radiometer observes from the surface, the dominant fractions of water in the middle and lower parts of such clouds have important influences on surface-observed radiances, which may make considering the water cloud phase reasonably valid in retrieval of cloud properties from surface observations for such conditions.*

L247 'Deep convective and nimbostratus clouds have COD values of greater than 23. ' But COD less than 20 seems to be included in Fig.4(c) (deep convection).

-->Thank you for an important comment. Please note that we considered 5x5 pixels of SGLI observation, which keeps a pixel including observation site at the center, while comparing with surface observation based cloud properties of ±30 minutes averages centering the SGLI overpass time. While comparing the results between SGLI and sky radiometer for different types of clouds, cloud type corresponding to the central pixel, which includes observation site,

is taken into account. As pointed out by the reviewer, there appears one sample having COD from SGLI less than 23 in Fig. 6(c). For this case, though the central pixel is a deep convective cloud, a part of cloud pixels around it is relatively thin clouds (anvil clouds) with COD less than 23.This resulted to have average COD less than 23 for one data sample in Fig. 6(c).

Taking this valuable comment into account, we clarified this ambiguity in the revised manuscript by providing additional descriptions in both text and figure caption (**Page 8, Lines 263-265; Page 9, Line 311-314; Page 9, Lines 324-325; Page 22, Caption of Figure 5; Page 23, Caption of Figure 7**)

**References**

Cahalan, R. F., Ridgway, W., Wiscombe, W. J., Bell, T. L. and Snider, J. B: The albedo of fractal stratocumulus clouds, Journal of the Atmospheric Sciences, 51, 2434-2455,10.1175/1520-0469(1994)051<2434:TAOFSC>2.0.CO;2, 1994.

Khatri, P., Iwabuchi, H., Hayasaka, T., Irie, H., Takamura, T., Yamazaki, A., Damiani, A., Letu, H., and Kai, Q.: Retrieval of cloud properties from spectral zenith radiances observed by sky radiometers, Atmospheric Measurement Techniques, 12, 6037-6047, 10.5194/amt-12-6037-2019, 2019.

King, N. J., Bower, K. N., Crosier, J., and Crawford, I.: Evaluating MODIS cloud retrievals with in situ observations from VOCALS-REx, Atmospheric Chemistry and Physics, 13, 191-209, 10.5194/acp-13-191-2013, 2013.

Li, Z., Zhao, F., Liu, J., Jiang, M., Zhao, C., and Cribb, M.: Opposite effects of absorbing aerosols on the retrievals of cloud optical depth from spaceborne and ground-based measurements, Journal of Geophysical Research: Atmospheres, 119, 5104-5114, 10.1002/2013jd021053, 2014.

Liu, J., Li, Z., Zheng, Y., Chiu, J. C., Zhao, F., Cadeddu, M., Weng, F., and Cribb, M.: Cloud optical and microphysical properties derived from ground-based and satellite sensors over a site in the Yangtze Delta region, Journal of Geophysical Research: Atmospheres, 118, 9141-9152, 10.1002/jgrd.50648, 2013.

Nakajima, T. and King, M. D.: Determination of the Optical Thickness and Effective Particle Radius of Clouds from Reflected Solar Radiation Measurements. Part I: Theory, Journal of the Atmospheric Sciences, 47, 1878–1893, https://doi.org/10.1175/1520-0469(1990)047<1878:DOTOTA>2.0.CO;2, 1990.

Nakajima, T., King, M. D, Spinhire, J. D., and Radke, L. F.: Determination of the Optical Thickness and Effective Particle Radius of Clouds from Reflected Solar Radiation Measurements. Part II: Marine Stratocumulus Observations, Journal of the Atmospheric Sciences, 48, 728–751, https://doi.org/10.1175/1520-0469(1991)048<0728:DOTOTA>2.0.CO;2, 1991.

Zhang, Z. and Platnick, S.: An assessment of differences between cloud effective particle radius retrievals for marine water clouds from three MODIS spectral bands, Journal of Geophysical Research, 116, 10.1029/2011jd016216, 2011.